# Carbothermal Synthesis of Ni/Fe Bimetallic Nanoparticles Embedded into Graphitized Carbon for Efficient Removal of Chlorophenol

**DOI:** 10.3390/nano11061417

**Published:** 2021-05-27

**Authors:** Min Zhuang, Wen Shi, Hui Wang, Liqiang Cui, Guixiang Quan, Jinlong Yan

**Affiliations:** 1School of the Environment and Safety Engineering, Jiangsu University, Zhenjiang 212013, China; z17802592726@163.com; 2School of Environmental Science and Engineering, Yancheng Institute of Technology, Yancheng 224051, China; sw18351278375@126.com (W.S.); cuiliqiang411@163.com (L.C.); qgx@ycit.cn (G.Q.)

**Keywords:** nanoscale zero-valent iron, graphitized carbon, reduction, adsorption, chlorophenol

## Abstract

The reactivity of nanoscale zero-valent iron is limited by surface passivation and particle agglomeration. Here, Ni/Fe bimetallic nanoparticles embedded into graphitized carbon (NiFe@GC) were prepared from Ni/Fe bimetallic complex through a carbothermal reduction treatment. The Ni/Fe nanoparticles were uniformly distributed in the GC matrix with controllable particle sizes, and NiFe@GC exhibited a larger specific surface area than unsupported nanoscale zero-valent iron/nickel (FeNi NPs). The XRD results revealed that Ni/Fe bimetallic nanoparticles embedded into graphitized carbon were protected from oxidization. The NiFe@GC performed excellently in 2,4,6-trichlorophenol (TCP) removal from an aqueous solution. The removal efficiency of TCP for NiFe@GC-50 was more than twice that of FeNi nanoparticles, and the removal efficiency of TCP increased from 78.5% to 94.1% when the Ni/Fe molar ratio increased from 0 to 50%. The removal efficiency of TCP by NiFe@GC-50 can maintain 76.8% after 10 days of aging, much higher than that of FeNi NPs (29.6%). The higher performance of NiFe@GC should be ascribed to the significant synergistic effect of the combination of NiFe bimetallic nanoparticles and GC. In the presence of Ni, atomic H* generated by zero-valent iron corrosion can accelerate TCP removal. The GC coated on the surface of Ni/Fe bimetallic nanoparticles can protect them from oxidation and deactivation.

## 1. Introduction

Recently, nanoscale zero-valent iron (nZVI) in pollutant remediation has attracted significant attention because nZVI can maximize the benefits of ZVI [1,2]. Owing to the nanoscale particle size, nZVI has a much larger specific surface as compared to ZVI, and more reactivity can be provided for targeted contaminant adsorption and reduction [3]. However, the following major problems inhibit the practical application of nZVI: (1) nZVI particles are prone to agglomeration due to the small particle size, large surface energy, and strong magnetic properties, resulting in decreased specific surface area and reactive sites [4]; (2) nZVI is prone to be easily oxidized under ambient environmental conditions, and the generated iron oxide passivation layer would inhibit the electron transfer [5]; (3) nZVI particles are easily lost in the water environment, making it difficult to recycle in the in situ contaminant remediation. Ensuring the high activity and longevity of nZVI is the key to its practical application, and it is also the challenge of the current technology.

In order to enhance the stability and anti-passivation properties of nZVI, great efforts have been focused on surface modification technology. Among them, doping with a second metal to form bimetallic nanoparticles is an efficient method to alleviate the formation of the passivation layer [6,7]. In the second metal-doped nZVI system, Fe^0^ acts as a reactive electron donor, while the second metals serve as the carrier and catalyst to adsorb H_2_ and convert it into activated atomic H* [8,9]. These doping metals, including Pd, Pt, Cu, Co, and Ni, not only enhance the reactivity of nZVI but also inhibit surface oxidization [10,11,12]. Although the catalytic hydrogenation activity of Ni is weaker than Pd and Pt, it is closer to having a practical application due to its low cost and good corrosion stability [6]. In addition, Fe/Ni bimetallic particles have a good removal effect on various pollutants in aqueous solutions [13]. For instance, Dong et al. found that Ni/Fe exhibited improved degradation efficiency towards tetracycline [6]. Tian et al. found that over 90% 1,1-trichloro-2,2′bis(p-chlorophenyl)ethane (DDT) was degraded by Ni/Fe, much higher than that of nZVI [14].

In practical applications, bimetallic Ni/Fe nanoparticles are prone to agglomeration due to their inherent magnetic force and van der Waals forces, which leads to their activity being easily reduced, and maintaining their stability remains a challenge [15,16]. Immobilizing Ni/Fe nanoparticles on supporters proved to be a potential solution to the above problems. Carbon materials display various unique physicochemical properties, such as large specific surfaces, high porosity, good conductivity, and stability, which can improve the dispersibility of Ni/Fe nanoparticles [17,18]. Ni/Fe nanoparticles are usually loaded onto carbon supports by a traditional liquid-phase reduction method using borohydride as a reducing agent after the iron salts are immersed in the carbon supports [13]. Though Ni/Fe nanoparticles are often loaded onto the surface of the carbon supports by the above method, nanoparticles cannot be separated well without space confinement and thus, the agglomeration cannot be effectively solved [4,19]. Besides, the bimetallic Ni/Fe nanoparticles on the surface of carbon supports can easily be exposed to the air or react with water, resulting in the formation of a passivation layer, which may reduce the durability of bimetallic Ni/Fe nanoparticles [20]. Therefore, improving the stability and durability of bimetallic Ni/Fe nanoparticles by optimizing synthesis methods is still challenging.

The carbothermal reduction method has been used to synthesize the nZVI/C composites because carbon is reductive at high temperatures [20,21]. It has been demonstrated that this method can improve the stability of nZVI in the air. The expensive and toxic borohydride can also be avoided in this method. Nonetheless, carbon-supported bimetallic nanoparticles synthesized using this method have been rarely reported. Motivated by this, a Ni/Fe-based complex was prepared and acted as the precursor for carbothermal reduction synthesis, and Ni/Fe embedded into graphitic carbon (NiFe@GC) was finally produced via one-step pyrolysis (Scheme 1). This synthesis has several advantages, as follows: (1) in comparison with the traditional metal salt impregnation method, a Ni/Fe-based complex, as a precursor, can better ensure the dispersion of metal ions; (2) Fe(Ni)-O clusters were isolated by ligands in the complex and then reduced in situ to bimetallic Ni/Fe nanoparticles, which can ensure the stability; (3) ligands were pyrolyzed to graphitic carbon, which can provide conductive and larger surface area support to immobile nanoparticles. The obtained NiFe@GC exhibited a large surface area, well-designed structure, and controlled particle size, which was further used for 2,4,6-trichlorophenol (TCP) removal. The objectives of this work were (1) to analyze the structure of the newly prepared NiFe@GC by transmission electron microscopy (TEM), N_2_ sorption, and X-ray diffraction (XRD); (2) to explore the reductive transformation of TCP by NiFe@GC; (3) to investigate the effects of various solution conditions (including pH, initial concentration, reaction temperature, dosage, and ionic strength) on TCP removal by NiFe@GC; (4) to elucidate the synergistic effect and removal mechanism of TCP. The results of this work will provide information for the structure design, surface optimization, and water remediation of nZVI-based materials.

## 2. Materials and Methods

### 2.1. Materials and Chemicals

This study mainly used the following chemical reagents: ethylenediaminetetraacetic acid (EDTA, ≥99%), N,N-dimethylformamide (≥99.8%, DMF), triethylamine (AR), iron nitrate nonahydrate (Fe(NO_3_)_3_·9H_2_O, AR), nickel nitrate hexahydrate (Ni(NO_3_)_2_·6H_2_O, AR), 2,4,6-trichlorophenol (TCP, ≥99%), hydrochloric acid (HCl, AR), potassium borohydride (KBH_4,_ AR), sodium hydroxide (NaOH, AR), Tert-butyl alcohol (≥99%, AR), and 1,10-Phenanthroline (98%, AR). All chemicals were purchased from Sinopharm Chemical Regent Co., Ltd. (Shanghai, China), and were used without further purification. All solutions were prepared in deionized water.

### 2.2. Preparation and Characterization of NiFe@GC, Fe@GC, and FeNi NPs

To synthesize NiFe@GC with different proportions by an organic–inorganic complex method, the operation steps were as follows: iron salt and nickel salt were dissolved in a certain molar ratio (Ni content: 10%, 30%, 50%) in DMF, named Solution A. A total of 0.584 g of EDTA and 3.5 mL of triethylamine were mixed in 30 mL of the DMF solution, and Solution B was obtained. Solutions A and B were mixed and stirred for several minutes, then centrifuged, washed with DMF several times, and then placed in a vacuum drying oven at 60 °C for drying to obtain a nickel/iron-EDTA complex. The nickel/iron-EDTA complex was then placed into a tube furnace and pyrolyzed at 800 °C in N_2_ atmosphere with a heating rate of 5 °C min^−1^, and the NiFe@GC-10, NiFe@GC-30, and NiFe@GC-50 were obtained after natural cooling to room temperature. The Fe@GC was prepared with the same process but without a Ni source. FeNi bimetallic nanoparticles (FeNi NPs), with a Ni addition amount of 50%, were prepared by the liquid-phase reduction method with KBH_4_ as the reducing agent, according to a previous report [13]. In order to investigate the TCP adsorption by GC, the GC was prepared by washing the NiFe@GC-50 with 1.0 M HCl.

TEM (JEM-2100F, JEOL, Tokyo, Japan) was used to analyze the morphology of the particles. XRD (Rigaku D/MAX-RB, Tokyo, Japan) was performed to check the possible crystallinity of the samples before and after use. X-ray photoelectron spectroscopy (XPS, Perkin–Elmer, Hopkinton, MA, USA) was used to analyze the elements content. The specific surface area and pore structure were analyzed by the N_2_ adsorption–desorption (ASAP 2020, Micromeritics, Norcross, GA, USA) isothermal and pore size distribution curve. The Brunauer–Emmett–Teller (BET) method was utilized to calculate the specific surface areas, and the pore size distributions were derived from the desorption branches of the isotherms using the Barrett–Joyner–Halenda (BJH) model.

### 2.3. Batch Experiments

Experiments of TCP removal by the above samples were conducted in sealed bottles full of TCP aqueous solution. The bottles were then placed in a temperature-controlled shaker with a speed of 200 rpm. Water samples were taken by a syringe at different time points and then filtered through a 0.22 μm filter membrane for the determination of the TCP concentration. The concentration of the solution was tested with an ultraviolet-visible spectrophotometer (TU 1810), and the wavelength was 294 nm. In addition, the effects of the NiFe@GC-50 dosage (0.5–1.0 g L^−1^), TCP concentration (10–200 mg L^−1^), solution pH (3–11), and reactive temperature (298.15–328.15 K) were investigated. The experiment conditions were kept constant: the NiFe@GC-50 dosage, TCP concentration, solution pH, and reactive temperature were 0.5 g L^−1^, 30 mg L^−1^, 3.0, and 298.15 K, respectively. When investigating a given factor, the other factors were kept constant. The effects of inorganic anions (SO_4_^2^^−^, HCO_3_^−^, NO_3_^−^) and HA on the removal efficiency of TCP by NiFe@GC-50 were studied under similar experimental conditions as described above; the concentration of inorganic anions (HA) ranged from 10–100 mM.

## 3. Results

The morphology of the NiFe@GC-50 and Ni/Fe nanoparticle distribution was characterized by the TEM. As shown in Figure 1a, the Ni/Fe nanoparticles were embedded into the carbon matrix after the pyrolysis of the Ni/Fe–EDTA complex at a high temperature. The particle size of the Ni/Fe nanoparticles was inhomogeneous; large particles with sizes between 50–100 nm and small particles with sizes below 20 nm were found. During the high-temperature calcination process, the carbon skeleton may have been destroyed, so larger particles were formed by the slight aggregation of finer particles. The core–shell structure is obvious, as seen from the high-resolution TEM (HRTEM) image in Figure 1b, indicating that the Ni/Fe nanoparticles were coated with carbon shell, and the carbon shell was highly graphitized due to the catalytic effect of Ni/Fe nanoparticles at a high temperature. Confining Ni/Fe nanoparticles in graphitic carbon can prevent particle agglomeration and oxidation in the application, while the highly graphitized shell will not hinder electron transference of Ni/Fe nanoparticles in the core [21].

The XRD patterns of NiFe@GC-50 and Fe@GC are presented in Figure 2a and the standard diffraction peaks are shown in Appendix A. The XRD pattern of Fe@GC clearly shows the presence of body-centered Fe^0^ (JCPDS 06-0696), with diffraction peaks of 44.5° and 65^o^, indicating the Fe species was reduced by the carbothermal reduction reaction at a high temperature. The NiFe@GC-50 shows different characteristic diffraction peaks with Fe@GC; the Fe^0^ phase disappeared and evolved to a new phase (Fe_0.64_Ni_0.36_, JCPDS 47-1405). The diffraction peaks at 2θ of 43.60°, 50.79°, and 74.68° belong to the (111), (200), and (220) planes of Fe_0.64_Ni_0.36_ [22]. The formation of Fe_0.64_Ni_0.36_ is due to the alloying of nickel and iron in the bimetallic complex under a high-temperature treatment. Importantly, no characteristic diffraction peak of iron oxide appeared on both the XRD patterns of Fe@GC and NiFe@GC-50. The FeNi NPs prepared from the liquid-phase reduction method show an obvious diffraction peak of iron oxide at 2θ = 34.2°, indicating that FeNi NPs were easily oxidized in the testing process (Appendix A). The XRD results indicate that Fe@GC and NiFe@GC-50 have better antioxidant abilities. According to the XPS result (Appendix A), the atomic content of Fe and Ni is 26.39 at% and 14.2 at%, and the Ni content is lower than the additional content, and this may be because part of the Ni did not participate in the coordination reaction with the EDTA and was then removed in the subsequent washing process.

The N_2_ adsorption–desorption isotherm and the pore size distribution of NiFe@GC-50 are shown in Figure 2b. The NiFe@GC-50 shows a type IV isotherm with an obvious hysteresis loop at P/P_0_ = 0.4–1.0, indicating the mesoporous structure [23]. The BET specific surface area was 252 m^2^ g^−1^, much larger than the reported FeNi NPs and supported FeNi NPs [18,24]. The pore size distribution curve in Figure 2b shows that the main size was centered at 4.0 nm, further indicating that the mesopore is dominant in NiFe@GC-50 structures. During the high-temperature pyrolysis of the NiFe–EDTA complex, various gases would blow out the carbon skeleton, resulting in many mesopores. The enlarged specific surface area will expose more active sites for TCP molecules, which is conducive to reactivity. The mesoporous structure will provide smooth channels between Ni/Fe active sites and TCP, which can facilitate a mass transfer.

In order to determine the effect of Ni doping on TCP removal, Fe@GC and NiFe@GC with different Ni/Fe ratios were used in the batch experiment, and the results are shown in Figure 3. After 180 min of reacting, the removal efficiencies were 81.6%, 87.9%, and 94.1% when the additional amounts of Ni were 10 wt%, 30 wt%, and 50 wt%, respectively. However, without Ni doping, the removal of TCP was only 78.5%. The results reveal that Ni doping can significantly improve the removal efficiency of TCP because galvanic cells are formed between coupled Fe^0^ and Ni^0^, resulting in a continuous electron flow and enhanced catalytic activity [7,24]. Here, the optimum Ni amount was 50 wt%, with the initial TCP concentration of 30 mg L^−1^ and dosage of 1.0 g L^−1^. There was no need to further increase the Ni content in the given reaction conditions. After all, excessive Ni is unfavorable from the economic and environmental perspectives. The TCP removal efficiency by GC was 38% (Appendix A). The porous structure of GC is beneficial to TCP accumulation, and the high degree of graphitization is conducive to the formation of π–π conjugated bonds between GC and TCP, which can enhance the accumulation of TCP on GC. In comparison to FeNi NPs prepared via the liquid reduction methods, the removal efficiency of TCP by NiFe@GC-50 was increased by 49.9%. In the NiFe@GC structure, the Ni/Fe nanoparticles were well-dispersed and embedded into the graphitic carbon matrix by using the metal ion–ligand chelation and carbothermal reduction method, which can not only increase the number of active sites but also prevent the formation of a passivation layer on the surface, leading to more efficient reactivity [25]. Hence, the cooperative system, including Fe^0^, Ni^0^, and GC, greatly enhanced the TCP removal.

In order to analyze whether the embedding structure would alleviate the surface passivation, the removal efficiency of TCP by FeNi NPs and NiFe@GC-50 after 2, 5, and 10 days of aging was tested, and the results are shown in Figure 4a. After aging, the removal efficiency of TCP by NiFe@GC dropped to 78.5% (2 days of aging) and 72.3% (10 days of aging). For FeNi NPs, after 2 and 10 days of aging, the removal efficiency dropped to 28.7% and 13.1%, respectively. As compared with fresh samples, the activity of NiFe@GC-50 can remain at 76.8%, while the activity of FeNi NPs can only remain at 29.6%, indicating that NiFe nanoparticles embedded in GC can alleviate the passivation of its surface and increase the reaction life.

In Figure 4b, the XRD analysis shows that the crystalline of NiFe@GC-50 was unchanged before and after aging, and there was no obvious characteristic peak of iron oxide after aging, indicating that the embedded structure can alleviate surface passivation. However, FeNi NPs also only have the peak of the iron oxide compound, indicating that FeNi NPs are prone to oxidation without a protective layer during the preparation and use process. The above results prove that the embedded structure can effectively protect FeNi bimetallic particles and inhibit the formation of an iron oxide passivation layer, thereby prolonging the service life of bimetallic nanoparticles [25].

The pH value of the solution will affect the redox process of nZVI-based materials; the effects of pH on TCP removal with NiFe@GC-50, Fe@GC, and FeNi NPs are depicted in Figure 5a. The NiFe@GC-50 exhibited higher TCP removal efficiency than Fe@GC and FeNi NPs under both acidic and alkaline conditions, which can be attributed to the effect of the embedded structure and Ni doping. When the pH increased from 3 to 11, the removal efficiency of TCP by NiFe@GC-50 decreased from 94.1% to 44.2%, which can be ascribed to the following factors. Under acidic conditions, there is a large amount of H^+^ in the aqueous solution, which will accelerate the corrosion of zero-valent iron and generate a large amount of H_2_. Ni has a strong adsorption capacity for H_2_ and can convert the adsorbed H_2_ into atomic H*, which is beneficial to the hydrodechlorination reaction and promotes the removal of TCP [26]. Moreover, H^+^ in an acidic reaction system can dissolve the passivation layer on the surface of NiFe@GC-50, which would accelerate the contact between TCP and NiFe@GC-50 [15]. When the pH value was further increased to 9 and 11, the removal efficiency of TCP by NiFe@GC-50 dropped to 59.8% and 44.2%, respectively, indicating a higher pH value here is unfavorable for TCP removal. Under alkaline conditions, OH^−^ in the solution can react with Fe^2+^ or Fe^3+^ to generate hydroxide precipitates covering the NiFe@GC surface, which will hinder the contact between TCP and NiFe@GC-50 [14]. However, when the pH increased from 3 to 11, the removal efficiency of TCP by FeNi NPs decreased from 44.2% to 6.9%. The TCP removal efficiency by NiFe@GC-50 remained at 46.9%, while FeNi NPs only remained at 15.6%. This further shows that surface passivation can be relieved when the NiFe nanoparticles are embedded into a GC matrix.

As shown in Figure 5b, the effect of the dosage of NiFe@GC-50, Fe@GC, and FeNi NPs nanoparticles on the removal of TCP was studied. With an increase in dosage, the removal efficiency of TCP increased due to the increase in active sites [27]. As the dosage of NiFe@GC-50 increased from 0.2 g L^−1^ to 1.0 g L^−1^, the removal efficiency of TCP increased from 44.2% to 94.1%, respectively, after a 180 min reaction time. When the dosage was further increased from 1.0 to 1.4 g L^−1^, the removal efficiency only increased to 97.2%, indicating that there were enough active sites for TCP at the dosage of 1.0 g L^−1^. Under any dosage conditions, the removal efficiency of NiFe@GC-50 for TCP was higher than that of Fe@GC and FeNi NPs, which further shows that the embedded structure and Ni doping can improve the reactivity.

The effect of the initial concentration of TCP was studied with different initial concentrations ranging from 10 mg L^−1^ to 200 mg L^−1^. As shown in Figure 5c, the TCP removal efficiency by NiFe@GC-50 was 100%, 94.1%, 85.2%, 78.9%, 67.4%, 66.4%, and 66.9% at the initial TCP concentrations of 10, 30, 50, 75, 100, 150, and 200 mg L^−1^, respectively. The removal efficiency of TCP is gradually decreased with the increase of initial concentration. This can be explained by the competitive adsorption mechanism. In fact, when the sample dose is constant, the amount of surface reactive sites is also fixed. As the initial concentration of TCP increases, the available reactive sites are gradually decreased due to the limited active site and adsorption area being constantly occupied [28]. In addition, as the reaction continues, NiFe@GC-50 will also generate a passivation layer, which will also result in a decrease in reactive sites and removal efficiency. However, it is worth noting that under any concentration conditions, the removal efficiency of NiFe@GC-50 for TCP is higher than the other two samples, indicating that the surface has more adsorptive and reductive sites.

As seen from Figure 5d, the HCO_3_^−^and NO_3_^−^ inhibit the removal of TCP by NiFe@GC-50, and the removal efficiency is decreased with the increase of the anion concentration. In a solution, CO_3_^2−^ ionized from HCO_3_^−^ can react with Fe^2+^, and the generated insoluble Fe salt will cover the surface of NiFe@GC-50, forming a passivation layer and hindering mass transfer and electron transfer, thereby reducing reactivity [29]. As oxidizing ions, NO_3_^−^ will compete with TCP for active sites on the Ni surface when it coexists with TCP in the same system, resulting in the decreased active sites for TCP. For SO_4_^2−^, it can enhance the removal of TCP at lower concentrations, which may be due to the dissolution of the passivation layer. HA has a high content in soil and groundwater and it plays an important role in both adsorption and electron transfer processes. When the HA with concentrations of 10, 50, and 100 mM was added to the 30 mg L^−1^ TCP solution, the removal efficiency of TCP by NiFe@GC-50 dropped to 59.8%, 69.1%, and 66.0%, respectively, indicating that HA has an adverse effect on TCP removal. Because HA is easily adsorbed on the surface of NiFe@GC-50, it will hinder the mass transfer and electron transfer and compete with TCP for active surface sites [30].

The removal of TCP by NiFe@GC-50 includes the adsorption and reduction process. Here, the adsorption behavior was further analyzed, and the Langmuir and Freundlich adsorption models were used to fit the TCP adsorption behavior by NiFe@GC-50, Fe@GC, and FeNi NPs. As shown in Figure 6b, the R^2^ of the Freundlich adsorption models for TCP removal behaviors by NiFe@GC-50 is 0.9239, which is larger than that of the Langmuir adsorption models (0.832, Figure 6a). Freundlich adsorption models can better describe the TCP removal behaviors by NiFe@GC-50, indicating that NiFe@GC-50 has a heterogeneous surface. The R^2^ values of the Langmuir models for TCP removal behaviors by Fe@GC and FeNi NPs are 0.9991 and 0.991, respectively, which are larger than those of the Freundlich adsorption models (0.855 and 0.9556), indicating that the TCP removal on the surface of Fe@GC and FeNi NPs is consistent with the single-layer adsorption [31].

The effect of the reaction temperature on TCP removal was also investigated, and the results are shown in Figure 7a. When the temperature increased from 298.15 K to 328.15 K, the removal efficiency of TCP by NiFe@GC-50 increased from 81.6% to 100% at a dosage of 0.5 g L^−1^, suggesting that the removal of TCP by NiFe@GC-50 is an endothermic reaction and a higher temperature is favorable for TCP transformation. The relationship between k_obs_ and T can be described by the Arrhenius equation: InK = −E_a_/RT + InA_0._

Where E_a_ (J·mol^−1^) is the activation energy, A_0_ (min^−1^) is the pre-factor, K (min^−1^) is the pseudo first-order adsorption rate constant, R (8.314 J·(mol·K)^−1^) is the gas constant, and T is the temperature at which the reaction proceeds [32]. The apparent activation energy E_a_ of NiFe@GC-50, Fe@GC, and FeNi NPs can be calculated from the slope of the straight line drawn by lnK and 1/T to 19.51 kJ·mol^−1^, 22.57 kJ·mol^−1^, and 26.88 kJ·mol^−1^, respectively (Figure 7b). As a catalyst, Ni metal can decrease the activation energy of TCP reduction reaction by NiFe@GC-50 [33]. Moreover, GC, as the supporting matrix, can improve the dispersibility of NiFe nanoparticles, and thus increase the surface-active sites and accelerate electron transfer in the catalytic reaction.

In order to analyze the effect of Fe^2+^, atomic H*, and Fe^0^ on TCP removal, a quenching experiment was conducted. Tert-butyl alcohol and 1,10-Phenanthroline were used to capture atomic H* and Fe^2+^, and the results are shown in Figure 8. The evident decrease of TCP removal can be found in the presence of scavengers, indicating that both atomic H* and Fe^2+^ were generated and participated in the TCP removal to a certain degree. Nevertheless, the removal of TCP still proceeded when the atomic H* and Fe^2+^ were captured, suggesting that the Fe^0^ also accounted for a certain degree of TCP removal. The phenomenon of the quenching experiment suggested that the synergistic impacts of Fe^2+^, atomic H*, and Fe^0^ co-induced TCP removal.

## 4. Conclusions

In this study, NiFe@GC was well-prepared by a one-step carbothermal reductive treatment of the bimetallic complex. Organic ligand over the precursors acted as the carbon source of graphitic carbon and the reductant for a bimetallic species. The graphitic carbon that simultaneously formed on the surface of bimetallic nanoparticles protected them from aggregation and oxidation. The obtained NiFe@GC-50 showed a remarkable removal performance of TCP with a high removal efficiency of 94.1%, which was much higher than that of Fe@GC and FeNi NPs. At the same time, the removal efficiency of TCP by NiFe@GC-50 after aging for 10 days was also higher than the other two materials and could be maintained at 76.8%. The activation energy of NiFe@GC-50 was 19.51 kJ·mol^−1^, lower than that of Fe@GC and NiFe NPs. These results demonstrate that the second metal (Ni) as an electron transfer medium and GC as a highly conductive protective shell could accelerate electron transfer from the Fe^0^ core to TCP and enhance their reactivity and longevity. Batch tests indicated that the acid solution and higher temperature were favorable for TCP removal. The adsorption process can be described by the Freundlich adsorption models. In a NiFe@GC system, atomic H* and Fe^2+^ may be the main reactive species for the dechlorination of TCP. In conclusion, NiFe@GC can be used as a competitive material for nZVI-based water treatment methods.

## Data Availability

The data presented in this study are available on the request from the corresponding author.

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
