# Peer review of "Carbothermal Synthesis of Ni/Fe Bimetallic Nanoparticles Embedded into Graphitized Carbon for Efficient Removal of Chlorophenol"

_nanomaterials, 2021, doi:10.3390/nano11061417_

Round 1

Reviewer 1 Report

The work presented in the manuscript is along the lines of previous work published by the researchers (Nanomaterials, 2019?). However, there appears to be on significant issue with the manuscript. Data for a control is plotted in Fig. 5(d), but details about the control are not addressed in the manuscript. Is the control simply carbon nanoparticles? This reviewer would like to see data for pure carbon since it will naturally absorb chemical species, such as TCP. I am confident that they have the data and can include it in the manuscript.

Finally, the quenching experiments are used to identify the mechanism behind the improved efficiency of NiFe@GC for removing TCP. The results in Fig. 8 are fine, but hardly provide evidence that Ni acts as an electron transfer medium. This is pure speculation and probably should be clearly stated as such. In the future, the authors might consider cyclovoltometry as a means for studying the redox processes of the system and thereby have concrete evidence about the oxidation states of the metals and how the alloy improves TCP removal.

Author Response

The work presented in the manuscript is along the lines of previous work published by the researchers (Nanomaterials, 2019?). However, there appears to be on significant issue with the manuscript. Data for a control is plotted in Fig. 5(d), but details about the control are not addressed in the manuscript. Is the control simply carbon nanoparticles? This reviewer would like to see data for pure carbon since it will naturally absorb chemical species, such as TCP. I am confident that they have the data and can include it in the manuscript.

Response: Thanks for your valuable comments. First of all, thank you for your attention to our work published in nanomaterials in 2019, but there are obvious differences between these two works, which are elaborated as follows:

  • In previous work, biomass-derived mesoporous carbon was synthesized by the hard template method, and nZVI was supported on its surface through liquid phased reduction with NaBH4 as reductant. In this work, Ni/Fe bimetallic nanoparticles embedded into graphitized carbon (NiFe@GC) were prepared from Ni/Fe bimetallic complex through carbothermal reductive treatment. It has been demonstrated that carbothermal reduction method can improve the stability of nZVI in air, and the expensive and toxic borohydride also can be avoided.
  • The surface modification methods of nZVI are different in these two works. In previous work, surface sulfidation was conducted to relieve surface oxidation of nZVI. In this work, nZVI was doped with a catalytic metal (Ni), and the bimetallic nanoparticles were formed. During the reduction of pollutants by nZVI, the catalytic metal serving as the medium receives electrons generated by Fe corrosion, later transferring them to the pollutants, which can control the surface passivation of nZVI.

In Figure 5d, the effect of co-existing ions on TCP removal was investigated. The detailed experiment condition was added, and described as “inorganic anions (SO42-, HCO3-, NO3-) and HA on the removal efficiency of TCP by NiFe@GC-50 were conducted under the similar experimental conditions as described above, the concentration of inorganic anions (HA) was ranging from 10-100 mM.” (Line 138-141, Page 4)

In order to investigate the TCP adsorption by GC, the GC was prepared by washing NiFe@GC by 1.0 M HCl. As shown in Figure S4, the TCP removal efficiency by GC is 38%. The porous structure of GC is beneficial to TCP accumulation, and the high degree of graphitization is conducive to the formation of π-π conjugate bonds between GC and TCP, which can enhance the accumulation of TCP on GC. (Line 117-118, Page 3; Line 200-202, Page 5)

Finally, the quenching experiments are used to identify the mechanism behind the improved efficiency of NiFe@GC for removing TCP. The results in Fig. 8 are fine, but hardly provide evidence that Ni acts as an electron transfer medium. This is pure speculation and probably should be clearly stated as such. In the future, the authors might consider cyclovoltometry as a means for studying the redox processes of the system and thereby have concrete evidence about the oxidation states of the metals and how the alloy improves TCP removal.

Response: Thanks for your valuable comments. Ni as a catalytic metal doped in nZVI has been widely investigated in previous reports ( The Science of the total environment, 715 (2020) 136822; Chemical Engineering Journal, 347 (2018) 669-681; Chemosphere, 266 (2021) 128976; Environmental Science & Technology, 52 (2018) 8627-8637). In the Ni doped nZVI system, Fe0 plays the role of electron donor and Ni0 plays the role of electron collector, and numerous galvanic cells thus are formed, which can promote the electron transfer between Fe0 and the target through a galvanic mechanism. Besides, Ni can facilitate the generation of activated atomic hydrogen (H) adsorbed on the bimetallic catalysts, which can also improve the reduction reactivity of nZVI. The related discussion is in Line 193-196.

Cyclic voltammetry can better analyze the reduction and oxidation reactions that occur on the surface of the electrode material in a certain solution system. However, the construction of the solution system and the exploration of the electrode forming process will take a certain amount of time. We will adopt your suggestions and use electrochemical methods to evaluate the surface redox process and electron transfer process of the bimetal system in future work.

Reviewer 2 Report

Authors study the TCP removal efficiency and the stability of NiFe nanoparticles embedded in graphitized carbon. The manuscript is well presented and discussed, materials are properly prepared and conclusions are drawn correctly.
Minor revisions are requested:
- authors should revise the English form. Some mispellings and many minor errors are present in the text, e.g. p.2 l. 58-60: sentence is not clear; l. 75: synthesis instead of reduction; l. 81: which can provide; and many others.

- p.4, l. 161-162: it would be better to provide the reader with reference or indication of the position of XRD peaks in Fe oxide. L. 162-164: sentence is not clear, english should be improved.

- Fig. 2: Misspelling in axis labels. Additionally the axis labels of inset of fig. 2b are not readable either for the size and the low resolution.

- p. 6, l. 204: the efficiency at 2 days aging of FeNi NPs is not 19.3% from the graph, it looks near to almost 30%. Please check and correct.

- l. 204-207: Which aging are the values in the sentence refering to?

- l. 212-219: authors correctly state that the embedded NiFe NPs are protected against passivation and therefore more active. However they also say here that FeNi NPs without embedding are less stable and get oxidized and passivated also during the reaction (and not only in preparation), explaining in this way the decrease of NPs activity. However it is clear from XRD that the FeNi NPs are from the beginning entirely oxidized, since no metal Fe phase is evident.  Are there other indication of the total oxidation of the NPs right after preparation? Are there references in literature? Do they refer to other surface passivation either than oxide?

- l. 237-238:the TCP removal rate of FeNi NPs at pH 11 is reported 15.6%, however from the graph the value seems under 10%. Please check and correct.

- Fig. 7b. Vertical axis label is not visible.

Author Response

Authors study the TCP removal efficiency and the stability of NiFe nanoparticles embedded in graphitized carbon. The manuscript is well presented and discussed, materials are properly prepared and conclusions are drawn correctly.
Minor revisions are requested:
- authors should revise the English form. Some mispellings and many minor errors are present in the text, e.g. p.2 l. 58-60: sentence is not clear; l. 75: synthesis instead of reduction; l. 81: which can provide; and many others.

Response: Thanks for your valuable comments.

Line 58-60: The sentence was revised as “Carbon materials displays various unique physicochemical properties such as large specific surface, high porosity, good conductivity and stability, which can improve the dispersibility of Ni/Fe nanoparticles.”

Line 75: “reduction” was replaced by “synthesis”

Line 81: “provided” was replaced by “provide”

- p.4, l. 161-162: it would be better to provide the reader with reference or indication of the position of XRD peaks in Fe oxide. L. 162-164: sentence is not clear, english should be improved.

Response: Thanks for your valuable comments. The stand diffraction peaks of iron oxide, Fe and Fe0.64Ni0.36 were provided in Figure S2. According to the stand diffraction peaks of iron oxide (PDF#40-1139), no characteristic diffraction peak of iron oxide is appeared on both the XRD patterns of Fe@GC and NiFe@GC, indicating that the embedded structure can prevent Fe0 from oxidation.

L162-164: The revised sentence is “The FeNi NPs prepared from the liquid-phase reduction method shows an obvious diffraction peak of iron oxide at 2θ=34.20, indicating that FeNi NPs were easily oxidized in the preparation or testing process.” (Line 168-169, Page 4) 

- Fig. 2: Misspelling in axis labels. Additionally the axis labels of inset of fig. 2b are not readable either for the size and the low resolution.

Response: Thanks for your valuable comments. We are very sorry for this error. The labels and size in Figure 2b have been revised. 

- p. 6, l. 204: the efficiency at 2 days aging of FeNi NPs is not 19.3% from the graph, it looks near to almost 30%. Please check and correct.

Response: Thanks for your valuable comments. We are very sorry for this error. The revised sentence is “For FeNi NPs, after 2 and 10 days of aging, the removal efficiency dropped to 28.7% and 13.1%.” (Line 216-218, Page 6)

- l. 204-207: Which aging are the values in the sentence refering to?

Response: Thanks for your valuable comments. The revised sentence is “After aging, the removal efficiency of TCP by NiFe@GC dropped to 78.5% (2 days aging) and 72.3% (10 days aging).” (Line 216-217, Page 6)

- l. 212-219: authors correctly state that the embedded NiFe NPs are protected against passivation and therefore more active. However they also say here that FeNi NPs without embedding are less stable and get oxidized and passivated also during the reaction (and not only in preparation), explaining in this way the decrease of NPs activity. However it is clear from XRD that the FeNi NPs are from the beginning entirely oxidized, since no metal Fe phase is evident.  Are there other indication of the total oxidation of the NPs right after preparation? Are there references in literature? Do they refer to other surface passivation either than oxide?

Response: Thanks for your valuable comments. We believe that FeNi NPs were oxidized during testing or transportation. The FeNi NPs we obtained are black and have been sealed in absolute ethanol. In the TCP reduction experiment, we took FeNi NPs out of ethanol, dried them under vacuum, and added them to the deoxygenated solution. After the above operation process, FeNi NPs still remain black. According to the previous reports, the FeNi NPs can be successfully prepared through the liquid phase reduction method used in our manuscript (Chemical Engineering Journal, 386 (2020) 123995; Chemosphere, 216 (2019) 499-506; Chemosphere, 160 (2016) 332-341; Environ Pollut, 227 (2017) 444-450). In our future work, we will further optimize the transportation or testing conditions to reduce the surface oxidation of FeNi NPs by air.

- l. 237-238:the TCP removal rate of FeNi NPs at pH 11 is reported 15.6%, however from the graph the value seems under 10%. Please check and correct.

Response: Thanks for your valuable comments. We are very sorry for this error. The removal efficiency by FeNi NPs at pH=11 is 6.9%. Here, we calculated the retention rate of TCP removal efficiency when the pH increased from 3 to 11. The revised sentence is “However, when the pH increased from 3 to 11, the removal efficiency of TCP by FeNi NPs decreased from 44.2% to 6.9%. The TCP removal efficiency by NiFe@GC-50 remained 46.9%, while FeNi NPs only remained 15.6%, respectively.”

- Fig. 7b. Vertical axis label is not visible.

Response: Thanks for your valuable comments. The error has been revised.

Reviewer 3 Report

This manuscript, nanomaterials-1197388 , reports very interesting results on the characteristics of "Ni/Fe Bimetallic Nanoparticles Embedded into Graphitized Carbon" .  However, major revision in several aspects is necessary as follows;

1) 3 different Ni/Fe bimetallic nanoparticles embedded into graphitized carbon were obtained through carbothermal treatment. However, only the ratios of Ni and Fe in the reactants for the bimetallic particles are given. The ratios of Ni and Fe in the bimetallic nanoparticles embedded in GC should be characterized to discuss the major experimental results in terms of the composition.

2) Except Figure 3, it is not clear which NiFeNP@GC was used in the experiments for Figures 1, 2, 4, 5, 6, 7, and 8.

3) It is recommended to suggest the mechanism of TCP removal by NiFe@GC nanoparticles by a scheme.

4) Superscript for the degree (angle) should be consistant in the manuscript.     

Author Response

This manuscript, nanomaterials-1197388, reports very interesting results on the characteristics of "Ni/Fe Bimetallic Nanoparticles Embedded into Graphitized Carbon" .  However, major revision in several aspects is necessary as follows;

  • 3 different Ni/Fe bimetallic nanoparticles embedded into graphitized carbon were obtained through carbothermal treatment. However, only the ratios of Ni and Fe in the reactants for the bimetallic particles are given. The ratios of Ni and Fe in the bimetallic nanoparticles embedded in GC should be characterized to discuss the major experimental results in terms of the composition.

Response: Thanks for your valuable comments. In the preparing process, the Ni content is 0%, 10%, 30% and 50%. The XPS was conducted to analysis the actual elements content of NiFe@GC-50, and the addition amount of nickel salt is 50% in the preparation.

According to the XPS result, the atomic content of Fe and Ni is 26.39 at% and 14.2 at%, and the Ni content is lower than the addition content, and this may be because part of Ni did not participate in the coordination reaction with EDTA and was removed in the subsequent washing process. The XPS survey is provided in Figure S3, and the related discussion is added in Line 171-175, Page 5.

  • Except Figure 3, it is not clear which NiFeNP@GC was used in the experiments for Figures 1, 2, 4, 5, 6, 7, and 8.

Response: Thanks for your valuable comments. The NiFe@GC respects NiFe@GC-50, which has been revised in the whole manuscript.

  • It is recommended to suggest the mechanism of TCP removal by NiFe@GC nanoparticles by a scheme.

Response: Thanks for your valuable comments. The GC-MS was conducted to analysis the reaction products, 2,4-dichlorophenol, 2,6-dichlorophenol, 2-chlorophenol, 4,-chlorophenol. We speculate that the possible dechlorination pathway is as following: the 2,4,6-TCP was first reduced to 2,6-dichlorophenol (2,4-dichlorophenol), then 2-chlorophenol (4,-chlorophenol). Because some products may be adsorbed by the NiFe@GC-50, and the detection results may not be very accurate.

In our future work, the analysis of dechlorination products still needs to be accurately analyzed by optimizing test conditions and dechlorination experiments.

  • Superscript for the degree (angle) should be consistant in the manuscript.   

Response: Thanks for your valuable comments. Superscript for the degree (angle) has been revised. Line 164, Page 4.
